# Multi-Locus Phylogenetic Analysis Revealed the Association of Six *Colletotrichum* Species with Anthracnose Disease of Coffee (*Coffea arabica* L.) in Saudi Arabia

**DOI:** 10.3390/jof9070705

**Published:** 2023-06-27

**Authors:** Khalid Alhudaib, Ahmed Mahmoud Ismail, Donato Magistà

**Affiliations:** 1Department of Arid Land Agriculture, College of Agricultural and Food Sciences, King Faisal University, Al-Ahsa 31982, Saudi Arabia; 2Pests and Plant Diseases Unit, College of Agricultural and Food Sciences, King Faisal University, Al-Ahsa 31982, Saudi Arabia; 3Department of Soil, Plant and Food Sciences, University of Bari A. Moro, 70126 Bari, Italy; donato.magista@gmail.com; 4Institute of Sciences of Food Production (ISPA), National Research Council (CNR), 70126 Bari, Italy

**Keywords:** anthracnose, coffee, *Colletotrichum*, multi-locus, phylogeny, pathogenicity

## Abstract

Several *Colletotrichum* species are able to cause anthracnose disease in coffee (*Coffea arabica* L.) and occur in all coffee production areas worldwide. A planned investigation of coffee plantations was carried out in Southwest Saudi Arabia in October, November, and December 2022. Various patterns of symptoms were observed in all 23 surveyed coffee plantations due to unknown causal agents. Isolation from symptomatic fresh samples was performed on a PDA medium supplemented with streptomycin sulfate (300 mg L^−1^) and copper hydroxide (42.5 mg L^−1^). Twenty-seven pure isolates of *Colletotrichum*-like fungi were obtained using a spore suspension method. The taxonomic placements of *Colletotrichum*-like fungi were performed based on the sequence dataset of multi-loci of internal transcribed spacer region rDNA (ITS), chitin synthase I (CHS-1), glyceraldehyde-3-phosphate dehydrogenase (GAPDH), actin (ACT), β-tubulin (TUB2), and partial mating type (Mat1–2) (ApMat) genes. The novel species are described in detail, including comprehensive morphological characteristics and colored illustrations. The pathogenicity of the isolated *Colletotrichum* species was assessed on detached coffee leaves as well as green and red fruit under laboratory conditions. The multi-locus phylogenetic analyses of the six-loci, ITS, ACT, CHS-1, TUB2, GAPDH and ApMat, revealed that 25 isolates were allocated within the *C. gloeosporioides* complex, while the remaining two isolates were assigned to the *C. boninense* complex. Six species were recognized, four of them, *C. aeschynomenes, C. siamense, C. phyllanthi,* and *C. karstii*, had been previously described. Based on molecular analyses and morphological examination comparisons, *C. saudianum* and *C. coffeae-arabicae* represent novel members within the *C. gloeosporioides* complex. Pathogenicity investigation confirmed that the *Colletotrichum* species could induce disease in coffee leaves as well as green and red fruits with variations. Based on the available literature and research, this is the first documentation for *C. aeschynomenes, C. siamense, C. karstii, C. phyllanthi, C. saudianum,* and *C. coffeae-arabicae* to cause anthracnose on coffee in Saudi Arabia.

## 1. Introduction

The genus *Coffea* is a member of the family *Rubiaceae* and is indigenous to the African continent, specifically Ethiopia [1]. Under this genus, there are two subgenera, *Coffea* and *Baracoffea*, which together comprise about 103 species [2]. Among all the species, the two most common and economically grown commercial species worldwide are *C. canephora* (Robusta) and *C. arabica* L. (Arabica). Historically, the coffee species could be traced to the Kaffa region of Ethiopia, and were later introduced to other parts of the world by traders from Yemen in the 15th century [1]. From a geographical perspective, Saudi Arabia is located in close proximity to Ethiopia, where the coffee cultivation and spread started a few centuries ago, especially in the southwest of the Arabian Peninsula (Yemen and the southwest of Saudi Arabia) [3]. Coffee is grown in Jazan, Al Baha, Asir, and Najran regions of Saudi Arabia. Based on the statistics of the Fyfa Development Authority (FDA, government organization), approximately 78,000 coffee trees are cultivated in Saudi Arabia, with 84% located in the Addayer district of the Jazan region. The annual coffee bean production from these trees in Saudi Arabia is estimated to be around 500 tons [4].

Coffee berry disease, or coffee anthracnose, is caused by several *Colletotrichum* species and is a widespread issue affecting coffee plants in production areas globally [5]. The disease was first reported in 1922 in Kenya [6,7], causing losses of up to 75% [1], which later spread quickly to Angola, Ethiopia, Malawi, Cameroon, Uganda, and Tanzania [1,8,9]. The causal agent responsible for causing that disease was known as C. *coffeanum* var. *virulans* [10]. Later on, pathogenicity and morphological investigations conducted by various authors between the 1960s and 1990s led to the reclassification of *C. coffeanum* var. *virulans* as *C. kahawae* [11]. Hindorf’s [12,13,14] studies on the *Colletotrichum* population within coffee resulted in the description of three distinct species occurring on coffee berries: *C. coffeanum*, *C. gloeosporioides*, and *C. acutatum*. Thus far, 68 strains of *Colletotrichum*, comprising 35 distinct species, seems to cause coffee berry disease [15], leading to total crop losses of 50–80% [16]. Among the *Colletotrichum* species causing coffee berry disease, *C. fructicola*, *C. siamense*, and *C. asianum* have been specifically reported in northern Thailand [16]. In Vietnam, *C. boninense*, *C. truncatum*, *C. acutatum*, *C. gloeosporioides*, *C. gigasporum*, *C. karstii*, *C. walleri*, and *C. vietnamense* have been identified [17,18], while *C. gigasporum*, *C. gloeosporioides*, *C. siamense*, *C. theobromicola*, and *C. karstii* were documented in Mexico [19]. In China, eight species of *C. karstii*, *C. ledongense*, *C. fructicola*, *C. endophytica*, *C. tropicale*, *C. siamense*, *C. gigasporum*, and *C. brevisporum* were associated with anthranconse symptoms on leaves and fruit [20].

From a taxonomic point of view, *Colletotrichum* genus is considered cryptic and has undergone numerous taxonomic investigations in recent years [18,20,21,22,23,24,25]. These investigations have relied mainly on the data of different molecular markers’ multi-locus sequence analyses, where morphological characters alone are often insufficient for delineating several species. The frequently used markers, comprising internal transcribed spacer region rDNA (ITS), chitin synthase I (CHS-1), calmodulin (CAL), actin (ACT), β-tubulin (TUB), glyceraldehyde-3-phosphate dehydrogenase (GAPDH), translation elongation factor 1- α (EF1α), and the large subunit of RNA polymerase II (RPB2) [8,20,26,27] have been demonstrated to be consistent for resolving the difficulties involved in identifying different species of the *Colletotrichum* genus. Additional molecular markers, such as APN2/MAT-IGS, GAP-IGS, and ApMat, were proposed as potential markers for delineating species of the *C. gloeosporioides* complex [20,28,29]. For separable *Colletotrichum* species complexes, some genomic markers like ApMat may be increasingly effective for certain species like those within the *gloeosporioides* complex. However, these markers may be less effective in distinguishing between species in other complexes. [30].

Regrettably, coffee trees in southwest Saudi Arabia are threatened primarily due to unknown fungal diseases and other potential pathogens. Considering the recently published work [31], limited information on fungi reported on coffee in Saudi Arabia is available. Keeping this in view, the current research is dedicated to monitor and subsequently characterize the *Colletotrichum* fungi accompanied with coffee trees, which could contribute to potential losses in the quantity and quality of coffee in Saudi Arabia. This study used a combination of phylogenetic analysis, morphological examination, and pathogenicity assessments to define and describe *Colletotrichum* species related to coffee trees in Saudi Arabia.

## 2. Materials and Methods

### 2.1. Sampling and Isolation

Coffee plantations were surveyed during October, November, and December 2022 in Jazan, Al Baha, Najran, and Asir regions (Table 1). Eighty-five vegetative samples from various tree parts, including fruits, leaves, and twigs, showing anthracnose symptoms were collected. Isolation from plant samples was made after surface disinfection through successive washing in 70 % ethanol for 30 s, followed by a 1 min wash in household bleach containing 1% NaOCl, and finally rinsed in distilled sterilized water and were dried using sterile filter paper [20]. Small pieces measuring 2–5 mm^2^, located between the infected and healthy tissues, were placed on potato dextrose agar medium (PDA) supplemented with streptomycin sulfate (300 mg/L^−1^) and copper hydroxide (42.5 mg/L^−1^) to inhibit bacterial and some fungal contamination [32]. Under dark conditions, the plates were incubated at 25 °C until the growth of fungi became visible. To obtain purified cultures, a hyphal tip was excised from the margins of the colonies that had developed from the tissue fragments and placed onto a new PDA medium. The new PDA medium was then incubated under the same conditions. Subsequently, single spore isolates were obtained using a spore suspension method [33].

### 2.2. Molecular Characterization

#### 2.2.1. DNA Extraction, PCR Amplification, and Sequencing

The total genomic DNA was obtained from the harvested fresh mycelium of 7-day old cultures of *Colletotrichum*-like isolates grown on a PDA medium using the Dellaporta protocol for genomic DNA isolation [34]. Six gene regions, comprising the 5.8S nuclear ribosomal gene with two flanking internal transcribed spacers (ITS), chitin synthase (CHS-1), actin (ACT), beta-tubulin (TUB2), glyceraldehyde-3-phosphate dehydrogenase (GAPDH), as well as partial mating type (Mat1–2) (ApMat) genes were amplified and sequenced. These gene regions were amplified with the primer pairs ITS1 + ITS4 for ITS [35], ACT-783R + ACT-512F for ACT *act* [36], T1 [37] + Bt2b [38] for TUB2, GDF + GDR for GAPDH [39], and AMF1 and AMR1 for ApMat [29], respectively. The primers that were utilized to amplify and sequence the DNA of *Colletotrichum* isolates in this study are shown in Table 2. The PCR reaction was carried out in 25 µL reaction volume, comprising 10 µL PCR Master Mix (amaR OnePCR, GeneDirex, Inc., Las Vegas, NV, USA), 1 µL of template DNA, 1.5 µL from each primer, and 11 µL of ddH_2_O. The PCR was carried out using a 2720 Thermal Cycler (Applied Biosystems, Foster City, CA, USA), and the amplification conditions for ITS, CHS-1, ACT, GAPDH, and TUB2 were identical to those outlined by Damm et al. [27]. For the ApMat gene, we followed the PCR amplification conditions outlined by Silva et al. [29]. The generated PCR products underwent bidirectional sequencing via Macrogen (Seoul, Republic of Korea) in accordance with the manufacturer’s guidelines.

#### 2.2.2. Phylogenetic Analyses

All obtained sequences underwent nucleotide BLAST search engine via the NCBI (https://www.ncbi.nlm.nih.gov/ (accessed on 22 February 2022)) to check the potential similarity with the closely related taxa. The new released sequences were aligned with the nucleotide sequences of reference strains of *Colletotrichum* (Table 3) belonging to the same complex retrieved from the NCBI GenBank database (http://www.ncbi.nlm.nih.gov (accessed on 28 February 2022)), based on recent publications [23,40,41,42]. The taxonomic identity of the strains was investigated by phylogenetic analysis of combined gene regions. For the *C. boninense* species complex, the ACT, ITS, TUB2, and CHS-1 were utilized, while ITS, ACT, CHS-1, TUB2, GAPDH, and ApMat combined gene regions were employed for the *C. gloesporioides* species complex. MEGA XI v.11.0.8 was utilized for trimming and concatenating the multi-sequence alignment. The *C. gloesporioides* complex alignment has 113 taxa with 2905 characters, 681 parsimony-informative, 1535 distinct patterns, 527 constant sites, and 1697 singleton sites. The *C. boninense* complex alignment has 36 taxa with 1448 characters, 478 distinct patterns, 224 parsimony-informative, 214 singleton sites, and 1010 constant sites. IQ-TREE multicore version 2.2.0 [43] was employed to calculate the best-fit evolution model based on BIC by ModelFinder [44] and to infer the phylogenetic tree Maximum likelihood (ML) relying upon 10,000 ultrafast bootstrap support replicates [45] on the partitioned dataset [46].

The combined partitioned dataset with adapted substitution models was subjected to Bayesian analysis using MrBayes v3.2.6 on Cipres Science Gateway (www.phylo.org) (accessed on 22 February 2022), adapted by the previously ModelFinder calculation. The analysis was conducted in duplicate using four Markov chain Monte Carlo (MCMC) chains for 10,000,000 generations, and random trees sampling for every 1000 generations. During the Bayesian analysis, a temperature value of 0.10 and a burn-in of 0.25 were used. The analysis was set to stop automatically once the split frequencies’ average standard deviation became less than 0.01. For the *C. boninense* complex, we used 1210 samples from two runs, each of which yielded 806 samples, from which 605 were selected for the final analysis. For the *C. gloesporioides* complex, we used 6894 samples from two runs, each of which yielded 4596 trees, from which 3447 were sampled. The ML and Bayesian phylogenetic trees were viewed in FigTree v. 1.4.4 (http://tree.bio.ed.ac.uk/software/figtree (accessed on 15 March 2022)). 

### 2.3. Morphological Characterization

Morphological characterization of *Colletotrichum* species was carried out as previously published [8,27]. For each characteristic isolate, the shape and sizes of 50 conidia were documented. In addition, the conidiophores, seta, and appressoria measurements were made for at least 30 at 100×magnification using Leica DM2500 LED light microscope with interference contrast (DIC). Appressoria was produced by dropping approximately 50 μL of conidial suspension on a glass slide, fixing the cover slip, and incubating for 5 days at 25 °C within a moist chamber. The results are presented as the minimum and maximum values along with the mean value ± its corresponding standard deviation (SD) for all measurements. Description and illustrations of novel species of *Colletotrichum* were deposited in MycoBank [47].

### 2.4. Pathogenicity Tests

Koch’s postulates were applied, and pathogenicity was carried out under controlled laboratory conditions on detached leaves and fruits of *Coffea arabica* [20]. Selected isolates representing six *Colletotrichum* species were first grown for 7 days on a PDA medium at 25 °C. Leaves and fruits that were of equal size and age and in good health were chosen for the inoculation process. Leaves and fruits were subjected to surface disinfection with household bleach (NaOCl 1%) for a 2 min period before washing in sterile distilled water and air-drying. To ensure the accuracy of the experiment, six replicates were carried out for each isolate. Each replicate involved three leaves and five fruits. The leaves were gently punctured at three points on the midrib’s upper surface utilizing a sterile needle tip. Coffee fruits were wounded by pinpricking the fruit wall to approximately 1 mm depth. Using the actively growing margins of each isolate, 5 mm of mycelium plug was extracted and positioned onto the wounded sites. The control leaves were subjected to inoculation using solely sterile PDA plugs. After inoculation, the leaves and fruits were then transferred into plastic boxes with lining of wetted paper towels to maintain high relative humidity. These were then incubated for 5–7 days at 25 °C, while being observed every day to detect the development of any symptoms. This experiment was repeated twice.

### 2.5. Data Analysis

Statistical analysis of variance [48] was achieved through employing SPSS 16.0 statistical package (SPSS Inc., Chicago, IL, USA) to delineate the mean size ± SD (standard deviation) of lesion diameters. Discrepancies in lesions diameters were documented after performing one-way-ANOVA at *p <* 0.05 and 95% confidence level. The mean of the measured values was compared utilizing the Least Significant Difference (LSD) test (*p <* 0.05).

## 3. Results

### 3.1. Symptoms Observation and Isolation

The coffee trees’ young leaves exhibited visible symptoms of anthracnose in the form of randomly scattered minor, irregular brown to black lesions. These lesions could expand and merge, leading to the formation of necrotic black patches (Figure 1A,B), which gave leaves a scorched appearance. The necrotic tissues were usually cracked forming holes on the leaf blade and finally detached from branches. On the twigs, black speaks initially starting from the apical portion and extended along the twig surface, leading to the death of the apical and lateral shoots (Figure 1C). Upon observing the semi-immersed fruiting structures (acervuli), orange masses of conidia were detected on the necrotic tissues that were released. Prominent, sunken dark decay lesions could extend deeply into the fruit, ultimately leading to the decay of fruit pulp of green and red berries (Figure 1D–F). In total, 27 *Colletotrichum*-like isolates were obtained; 18 from leaves, 6 from fruit, and 3 from branches (Appendix A). The phylogenetic study comprised all the isolates obtained.

### 3.2. Molecular Characterization

The identification of all *Colletotrichum*-like isolates began with their classification up to the genus level, which relies upon their ITS sequences. Identity of isolates was further confirmed at the species level, based on the multi-locus phylogenetic analysis of the six-loci (ITS, ACT, CHS-1, TUB2, ApMat, and GAPDH) for our 27 sequences of *Colletotrichum* isolates along with reference sequences retrieved from GenBank (Table 3). This analysis revealed that 27 isolates were assigned into two species complexes, the *C. gloeosporioides* complex and *C. boninense* complex. Among the 27 isolates, 25 allocated within the *C. gloeosporioides* complex, and the remaining two belonged to the *C. boninense* complex. In the phylogenetic tree (Figure 2) of the six-loci ITS, ACT, CHS-1, TUB2, ApMat, and GAPDH, 25 isolates within the *C. gloeosporioides* complex clustered in four clades, eight of them with *C. siamense* and single isolate with *C. aeschynomenes*. Furthermore, two discrete clades were positioned far apart from all recognized species within the complex, and thus, they were recognized as new species and named *C. saudianum* and *C. coffeae-arabicae* (Figure 2). In the *C. boninense* complex phylogenetic tree (Figure 3), each of the two isolates were grouped in distinct clade. The phylogenetic analysis strongly supported the placement of PPDU41K in a clade with CBS129833, VPRI43652, and CBS126532 of *C. karstii*, as indicated by the high BS/BPP values (100%/1.0). This clade was recognized as *C. karstii* on the phylogenetic tree (Figure 3). The second isolate, PPDU36S, was grouped with the isolate CBS175.67 of *C. phyllanthi* within a clade highly supported with BS/BPP values (90%/1.0). Therefore, PPDU36S was identified as the known species *C. phyllanthi*.

### 3.3. Taxonomy

The morphological characteristics and multi-locus phylogeny helped designate the 27 isolates attained in this study into six distinct species. Four species, *C. aeschynomenes*, *C. siamense*, *C. karstii*, and *C. phyllanthi*, were firstly documented from coffee in Saudi Arabia, and a further two species were newly described.

***Colletotrichum saudianum*** Alhudaib and A.M. Ismail., sp. nov. MycoBank 848994; Figure 4.

Etymology: The name refers to the country of origin, Saudi Arabia.

Sexual morph not observed. Asexual morph on PDA. Conidiomata acervular, semi-immersed or superficial, globose, black, solitary, or gregarious, oozing white or buff conidial masses. Setae and chlamydospores not observed. Conidiophores hyaline, thin-walled, smooth, 1–3 branched, 1–2 septate. Conidiogenous cells hyaline, thin-walled, smooth, cylindrical to inflated at the base, 13.5–19.2 × 1.9–4.1 μm, mean ± SD = 15.3 ± 3.2 × 3.1 ± 0.57 μm. Conidia hyaline, thin-walled, smooth, aseptate, cylindrical to oblong, granular contents, and small guttules, rounded at apex, slightly obtuse at base, 11.6–14.5 × 3.9–5.2 mean ± SD = 12.8 ± 0.93 × 4.5 ± 0.38 μm, L/W ratio = 2.8. Appressoria dark brown, irregular in shape, sometimes roundish with undulate margins, 7.1–9.7 × 5.1–7.3 µm, mean ± SD = 7.9 ± 0.85 × 5.8 ± 0.65 µm, L/W ratio = 1.3.

Culture characteristics: the colonies grown on PDA were sparse and dense, with effuse mycelium mats that were initially white and became olivaceous buff to greenish olivaceous on the upper surface. On the reverse side, the colonies had iron grey to olivaceous grey color. The color darkened with age. Following 10 days of dark incubation at 25 °C, the colonies grown to the Petri plate edge, measuring 85 mm. Conidia were observed as orange masses released from semi-immersed acervuli.

Materials examined: SAUDI ARABIA, Asir Region, from leaves of *Coffea arabica* (Rubiaceae), 17 November 2022, A.M. Ismail, culture ex-type PPDU38H (holotype KSA-38H-2023); from leaves of *Coffea arabica* (Rubiaceae), 17 November 2022, A.M. Ismail (PPDU38B). Additional examined materials: SAUDI ARABIA, Al Baha Region from leaves lesions of *Coffea arabica* (Rubiaceae), 14 September 2022, A.M. Ismail (PPDU31M); SAUDI ARABIA, Jazan Region from fruit lesions of *Coffea arabica* (Rubiaceae), 13 October 2022 (PPDU28E).

Notes: According to the multi-locus phylogenetic analysis of the combined six genes, ITS, ACT, TUB2, CHS-1, GAPDH, and ApMat, 12 strains of *C. saudianum* formed an independent clade in the *gloeosporioides* complex (Figure 2). *Colletotrichum saudianum* is discerned from all species of the genus based upon its morphology, as it produces short conidia (mean ± SD = 12.8 ± 0.93 × 4.5 ± 0.38 μm) compared to those of *C. tainanense* (16–22 × 4.5–5 μm) [23], and *C. salsolae* (av. 15.3 × 5.8 μm) [8]. Furthermore, the conidia shape of *C. saudianum* is cylindrical, while those of *C. salsolae* are subglobose to long cylindrical. In addition, the conidiogenous cells of *C. salsolae* are wider (4–6.5 μm) than those of *C. saudianum* (1.9–4.1 μm). Furthermore, a BLASTn searching on the NCBI GenBank utilizing the ex-type strain PPDU38H’ ITS sequences revealed the closest matches to be 100% *C. gloeosporioides* (GenBank JX902431), 99.8% *C. aenigma* (GenBank OQ184880), and 99.8% *C. siamense* (GenBank OQ184036). In contrast, based on the ACT sequence, the closest matches found were 99.5% *Colletotrichum* sp. (GenBank KC790648) and 99% with *C. siamense* (GenBank OQ023904 and OQ023903). BLASTn search using TUB2 sequence yielded closest matches 100 % with *C. siamense* (GenBank MF143931), 99% with *C. salsolae* (GenBank MN746330), and 99% with *C. fructicola* (GenBank OP660827). However, the closest similarities using the CHS-1 sequence were 100% *C. gloeosporioides* (GenBank MF554932), 100% *Colletotrichum* sp. (GenBank KF451982), and 100% with *C. fructicola* (GenBank OQ702521). Based on the GAPDH sequence, the closest matches found were 95.7 % *C. siamense* (GenBank MF110883, MF110873) and 95.7% *C. dianesei* (GenBank KX094166). Additionally, the closest matches of the ApMat were 99.8% *Colletotrichum* sp. (GenBank KC790698), 97.4% *C. siamense* (GenBank OM816816, OM816807). The morphological comparisons and molecular analyses confirm that *C. saudianum* denotes a novel species within the *C. gloeosporioides* complex.

***Colletotrichum coffeae-arabicae*** Alhudaib and A.M. Ismail., sp. nov. MycoBank 848995; Figure 5.

Etymology: The name refers to the host plant (*Coffea Arabica*) from where the fungus was originally collected.

Sexual morph not observed. Asexual morph on PDA. Conidiomata are mostly solitary or in aggregates, semi-immersed in the mycelium, oozing orange masses of conidia. Setae are light to dark brown, thick-walled, mostly straight or slightly flexuous, cylindrical, sometimes inflated in the middle, slightly inflated or conical at the base, acute to slightly rounded at the tip, 2–3 septate, 40–118 × 3–5 μm. Conidiophores are hyaline, thin-walled, smooth, 2–4 branched, and 1–2 septate. Conidiogenous cells are hyaline, thin-walled, smooth, cylindrical to swollen, 13–24 × 3–6 μm, mean ± SD = 19 ± 3.2 × 5 ± 1 μm. Conidia hyaline, thin-walled, smooth, cylindrical to ellipsoid, aseptate, somewhat constricted at the middle, guttulate with some small guttules, rounded at apex, obtuse at base, 15.5–18.7 × 5.8–7.4 μm, mean ± SD = 17.3 ± 0.7 × 6.4 ± 0.5 μm, L/W ratio = 2.7. Appressoria medium to dark brown, thick-walled, irregular in shape, but often elliptical shaped, 6.9–11.8 × 4.6–7.8 µm, mean ± SD = 8.6 ± 1.56 × 6.1 ± 0.96 µm, L/W ratio = 1.4.

Culture characteristics: the colonies on PDA are fluffy with white raised cottony mycelia, turned dark mouse-grey in the center, pale grey with an entire margin. The reverse of the colonies is iron grey to olivaceous grey. Following a 7-day incubation at 25 °C in the dark, the colonies grown to the Petri plate edge, measuring 85 mm. The conidia appear as pinkish-orange masses released from semi-immersed acervuli.

Materials examined: SAUDI ARABIA, Jazan Region, from leaves of *Coffea arabica* (Rubiaceae), 12 October 2022, A.M. Ismail, culture ex-type PPDU26B (holotype KSA-26B-2023); from branches and leaves lesions of *Coffea arabica* (Rubiaceae), 12 October 2022, A.M. Ismail (PPDU27D, PPDU29F).

Notes: The *C. gloeosporioides* species complex is characterized by cylindrical conidia that have rounded ends and taper slightly towards the base, which is similar to the conidial morphology observed in *C. coffeae-arabicae* [8,25]. However, the multi-locus phylogenetic analysis revealed that the four *C. coffeae-arabicae* strains formed a discrete clade and were phylogenetically distinct from the current recognized species within the *gloeosporioides* complex. Furthermore, BLASTn search of the ex-type strain PPDU26B of *C. coffeae-arabicae* sequences revealed a variable sequence resemblance with other sequences within the NCBI GenBank from different species. The closest matches using the ITS had a 100% similarity to *C. siamense* (GenBank MT450691, MT450690, and MT450689). Furthermore, the closest ACT sequence match showed 100% similarity to *C. aenigma* (GenBank OQ698783 and OQ698782) and 100% to *C. siamense* (OQ698755). However, TUB2 showed the highest similarity 100% to *C. siamense* (GenBank OP660847; OP660836 and OP660829). However, the CHS-1 sequence revealed homology of 99.5% to *C. gloeosporioides* (GenBank MF554932 and ON723793) and 99% to *C. fructicola* (GenBank OQ703570). Moreover, the GAPDH sequences demonstrated 100% to *C. siamense* (GenBank MF110865; MN228537 and MN228536). Additionally, the ApMat sequences had 96.7% similarity with *C. siamense* (GenBank KX578771), 96.3 % with *C. siamense* (GenBank MW557490), and 96.1% with *C. siamense* (GenBank OM816816). The morphological comparisons and phylogenetic analyses ascribed *C. coffeae-arabicae* as a novel taxon within the *C. gloeosporioides* complex.

### 3.4. Pathogenicity Tests

Pathogenicity test results demonstrated that all the tested *Colletotrichum* isolates were able to induce disease symptoms similar to that recognized in the field on coffee leaves and fruits (Figure 6 and Figure 7). After 5 days, small brown lesions appeared nearby the inoculation site, which then grew and developed into large necrotic brown lesions with black margins (Figure 7A–D). Orange conidial masses have been recognized on the surface of necrotic lesions on leaves as well as on red fruit after 12 days (Figure 7D,F). No symptoms developed on the control leaves and fruits. The tested isolates of *C. saudianum* and *C. siamense* developed lesions 3 days earlier than the two isolates of *C. karstii* and *C. phyllanthi*, which developed lesions after 8 days. The LSD test revealed significant (*p <* 0.05) differences in lesion diameter induced by the tested isolates, of which *C. saudianum* PPDU38H caused the largest lesion diameter (1.63 cm), followed by *C. saudianum* PPDU28E, which produced lesion that reached 1.48 cm. Conversely, the remaining *Colletotrichum* isolates produced lesions that insignificantly (*p <* 0.05) varied in size from each other (Figure 6A). The majority of isolates produced larger lesion sizes on red fruit than green ones (Figure 7E, F), with the largest lesions caused by *C. siamense* PPDU27M (1.8 cm), *C. saudianum* PPDU38H (1.68 cm), *C. saudianum* PPDU28E (1.5 cm), and *C. coffeae-arabicae* PPDU29F (1.48 cm). In contrast, the smallest lesion sizes were caused by isolates *C. aeschynomenes* PPDU28A (0.88 cm), *C. siamense* PPDU40G (0.8 cm), *C. karstii* PPDU41K (0.5 mm), and *C. phyllanthi* PPDU36S (0.4 cm). On the other hand, the two isolates *C. coffeae-arabicae* PPDU29F and *C. saudianum* PPDU38H showed equal virulence on green fruit by producing similar lesion lengths (0.93, 0.9 cm, respectively), which were significantly (*p <* 0.05) larger than those of other isolates (Figure 6B). Contrariwise, both *C. karstii* PPDU41K and *C. phyllanthi* PPDU36S revealed much lowered lesion expansion rate around the inoculation site over the experimental progress either on leaves or green as well as red fruits (Figure 6A,B and Figure 7). The differences in lesion diameters among *Colletotrichum* species and even isolates of the same species attributed to their geographical origin or the plat part where they were isolated. It was also observed that mature fruits were more sensitive than green ones and exhibited larger lesions diameters. The artificial inoculation of *Colletotrichum* species onto detached coffee leaves and fruits resulted in the successful recovery of the fungi, fulfilling Koch’s postulates.

## 4. Discussion

*Colletotrichum* is a genus that comprises economically significant pathogenic species with numerous host plants worldwide. Few efforts have been made to assess the disease problems of *Coffea arabica* in Saudi Arabia. Therefore, this study represents the initial attempt to evaluate the occurrence and the diversity of *Colletotrichum* species that are linked to different symptom patterns recognized in coffee trees. During a planned survey carried out in October, November, and December 2022, various patterns of symptoms were observed in all 23 surveyed coffee plantations due to unknown causal agents. The well-known anthracnose symptoms were often observed on the leaves as minute black to dark brown lesions with asymmetrical margins. Infections on the twigs and branches typically start from the apical portion along the twig surface, leading to the death of the apical and lateral shoots. Green and red berries exhibited dark, sunken, prominent lesions that deeply extended into the fruit, causing the fruit pulp to decay. These observed symptoms coincided with those previously reported [19,49].

Accurate delineation of the causal organisms responsible for *Colletotrichum* infections is crucial, given the significant economic losses experienced by coffee plantations and the restricted knowledge of growers in this regard. In the present study, the ITS sequence data aided in placing the 27 isolates in the *C. gloeosporioides* and *C. boninense* species complexes, approving the usefulness of ITS sequencing for categorizing *Colletotrichum* isolates [24,50]. Furthermore, extensive phylogenetic inference depending upon multi-locus analyses of ITS, ACT, TUB2, CHS-1 GAPDH, and ApMat provided a firm resolution and allocated all *Colletotrichum* isolates associated with *Coffea arabica* into two distinct species complexes and additionally ascribed them into six species. Among the six species identified, four were already known, *C. siamense*, *C. aeschynomenes*, *C. karstii*, and *C. phyllanthi*, while two novel species, *C. saudianum* and *C. coffeae-arabicae*, were also identified. It was not easy to discriminate species of *C. gloeosporioides* complex depending upon the data of the five loci including, ITS, ACT, CHS-1, TUB2, and GAPDH. Interestingly, relying on the sequence data of the single gene ApMat adequately provided a robust separation between the species of the *C. gloeosporioides* complex, and the resulting tree has topology resembling the tree obtained by the six loci. It also aided in the confirmation of the identity of two newly described species in this study, namely *C. saudianum* and *C. coffeae-arabicae*. Our results are supported by those published by de Silva et al. [29], who confirmed that the ApMat marker solely was ultimately useful in disentangling species of the *C. gloeosporioides* complex isolated from *C. arabica* and other coffee species. Other studies have confirmed these findings. For example, Liu et al. [41] verified that the ApMat marker, along with GS, offers significant phylogenetic information and successfully separated 22 species in the *C. gloeosporioides* complex when compared to other used loci ITS, ACT, CHS-1, TUB2, GS, and GAPDH. In addition, the research of Khodadadi et al. [24] revealed that the ApMat, when combined with ITS and TUB2, could efficiently allocate the new species *C. noveboracense* to a discrete clade that was highly supported with Bayesian posterior probability and bootstrap values. Crouch et al. [51] first introduced the Apn2-Mat1 locus for differentiating species in the *C. graminicola* complex. This ApMat marker was subsequently used to separate species in the *C. gloeosporioides* complex [28,52,53,54]. Both GAPDH and TUB2 markers are widely considered highly effective barcodes for most *Colletotrichum* complexes and are widely used. However, complex-specific barcodes must still be utilized in conjunction with them to achieve accurate species delimitation [8,28,29]. In our case study, GAPDH and TUB2 sequence did not consistently delineate species within the cryptic species of *gloeosporioides* complex. Accordingly, using ApMat sequence data approved the affordability and reliability of this marker for differentiating species of *C. gloeosporioides* complex. Therefore, we recommend combining ApMat with other markers as a sufficient technique for classifying species within the *C. gloeosporioides* complex.

Based on the results of this study, the most frequently reported species belonging to the *C. gloeosporioides* complex were *C. siamense*, *C. aeschynomenes*, *C. saudianum*, and *C. coffeae-arabicae*. Only two isolates representing two species, *C. karstii* and *C. phyllanthi*, belonged to the *C. boninense* species complex, and these were separated at much lowered frequency (one isolate for each). Among the species of *C. gloeosporioides* isolated from coffee, *Colletotrichum saudianum* (12 isolates) was the most frequently isolated, followed by *C. siamense* (8 isolates) and *C. coffeae-arabicae* (4 isolates). In contrast, only a single isolate of *C. aeschynomenes* was recovered. The presence of six species of *Colletotrichum* associated with anthracnose disease on coffee indicates that more than one *Colletotrichum* species can colonize a single host, which is consistent with the conclusion of previous studies [16,19,25,27,55]. The compositions of *Colletotrichum* species from coffee appeared to differ according to the geographical origin, host, and species complex. For example, *C. kahawae* also appears to be host-specific to *Coffea* species and geographically restricted and widespread in the African continent or in low altitudes [8,11,15]. However, *C. kahawae* has been reported to cause anthracnose disease on different hosts in Australia, Europe, South Africa, and USA [8,56]. Furthermore, other members of the *C. gloeosporioides* complex, such as *C. siamense* and *C. fructicola*, are widely reported in coffee in several countries and are known to have a broader host range. Although several species have been reported to cause infection in coffee, the association of *C. aeschynomenes* and *C. phyllanthi* and the newly described species *C. saudianum* and *C. coffeae-arabicae* is considered the first report in Saudi Arabia and worldwide. The low incidence of *C. karstii* and *C. phyllanthi* and the fact that the only two isolates of these species induced the smallest lesions on coffee leaves and fruit indicate that these species are of little importance and do not contribute significantly to anthracnose disease. Previous studies have reported that *Colletotrichum karstii* is a causal agent of anthracnose disease on coffee in Vietnam and Mexico, but in low frequencies [18,19,20], which supports our results. *Colletotrichum phyllanthi*, on the other hand, has not been previously reported on coffee, and we report for the first time its association with anthracnose symptoms.

Koch’s postulates were fulfilled, indicating that all isolates were pathogenic to detached coffee leaves as well as green and red fruit with significant *p* < 0.05 variations in infection degree. Variations were also among isolates of the same species, with the most virulent species being *C. saudianum*, *C. siamense*, and *C. coffeae-arabicae*, which frequently recovered from coffee. On the other hand, the lowest dominant species, *C. aeschynomenes*, *C. karstii,* and *C. phyllanthi*, provoked the smallest lesions either on detached leaves or on fruit (Figure 6). According to the statistical analysis, there were significant differences between isolates. These differences could be attributed to the geographical origin of isolates or/and plant part where it was isolated. The leaf lesions caused by the six *Colletotrichum* species were similar; however, the symptoms development and lesion sizes varied among species. For example, leaves and fruit inoculated with *C. saudianum* and *C. siamense* developed lesions 5 days earlier and larger than the other species, whereas the two isolates of *C. karstii* and *C. phyllanthi* developed lesions after 8 days. Similar results were also reported, in which the *C. siamense* was faster in developing lesions on coffee leaves and *C. karstii* was the slowest species, which produced lesions after 30 days of inoculation [19]. Additionally, Cao et al. [20] found out that among tested *Colletotrichum* species; *C. siamense*, *C. gigasporum* and *C. karstii* were the most virulent on both Arabica and Robusta coffee red fruits and recorded the same infection incidence 100 %. While on green fruit, the infection incidence was lower and registered 50, 0, and 25 %, respectively. Moreover, Nguyen et al., [57], indicated that *C. fructicola* and *C. siamense* can induce lesions on detached green berries after inoculation; however, the efficacious infection rate was low. In a similar study, Prihastuti et al. [16] demonstrated that *C. fructicola* was the most virulent species in producing higher infection percentage (89.93 %) on red fruit than *C. asianum* (63.06%) and *C. siamense* (50.19%). Similarly, Waller et al., [11] indicated that *C. gloeosporioides* isolates from coffee are capable of causing disease only on ripe berries, leaves, and are not able to cause the infection of green berries. These findings were also confirmed in laboratory trials in Papua New Guinea, of which *C. gloeosporioides* only infected ripe red berries [58]. These results supported our findings, of which the red fruits were more severely affected than green ones. The reasons behind this could be the onset of senescence, which are characterized by reduced defensive systems, weakened tissues, and increased ethylene production.

## 5. Conclusions

Understanding the taxonomy and the pathogenicity of *Colletotrichum* is fundamental in coffee production regions in order to manage this economically important disease and secure the profitability of the coffee industry in Saudi Arabia. Knowing the distribution of *Colletotrichum* species could help to propose a suitable control program based on their sensitivity to fungicides. In this study, ITS, TUB2, ACT, CHS-1 were sufficient to distinguish *C. karstii* and *C. phyllanthi* within the *C. boninense* complexes. In contrast, ITS, TUB2, ACT, CHS-1, GADPH, and ApMat regions were fundamental to differentiate species within the *C. gloeosporioides* complex. Therefore, using GADPH and ApMat gene regions confirmed the reliability and affordability of these markers to differentiate between species of *C. gloeosporioides* complex. Although *C. siamense* has been previously reported on *Coffea arabica* and many host species, this is the first report of *C. siamense* causing anthracnose on coffee in Saudi Arabia. This was also the first report of *C. aeschynomenes* on coffee in Saudi Arabia and worldwide. In addition, the two novel species; *C. saudianum* and *C. coffeae-arabicae* were new additions to the *Colletotrichum* species causing anthracnose on coffee in Saudi Arabia and worldwide. Furthermore, the dominance of *C. saudianum* makes it an appropriate model for addressing questions of population structure and dispersal at broad geographical and landscape level. Hence, additional collections from coffee growing regions across the southwest of Saudi Arabia would therefore aid us characterize the population structure of this important pathogen and to confirm whether this species is indeed the dominant *Colletotrichum* species.

## Figures and Tables

**Figure 1 jof-09-00705-f001:**
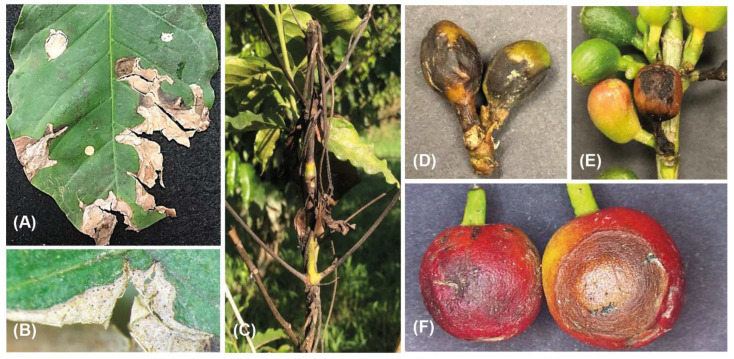
Anthracnose symptoms detected in the surveyed coffee plantations. Small lesions with irregular margins merging to develop large necrotic black patches starting from the leaf margins and moving to the middle of the leaf blade (**A**); close focus on the necrotic area showing the semi-immersed acervuli (**B**); dark necrotic patches result in the death of both the lateral and apical shoots (**C**); dark to brown sunken and depressed lesion on the green and red fruit berries (**D**–**F**).

**Figure 2 jof-09-00705-f002:**
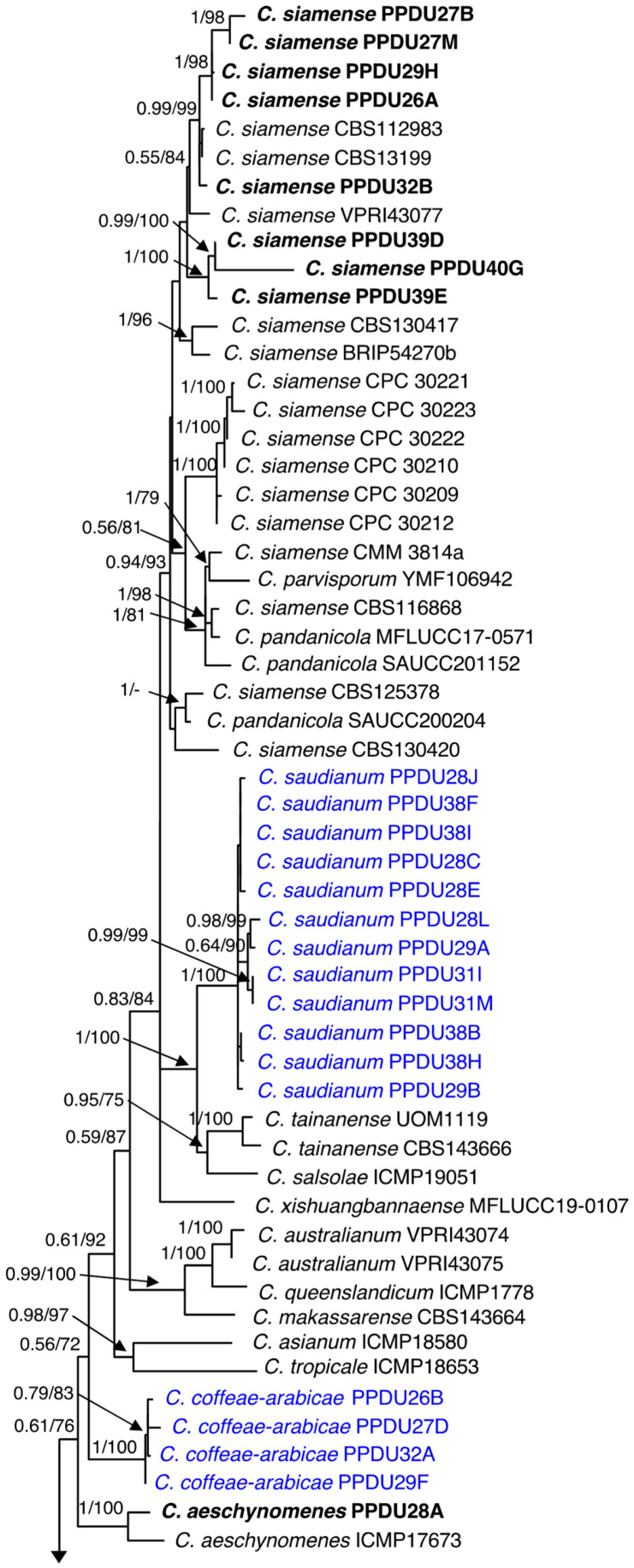
Maximum likelihood tree obtained through heuristic searches of the six-loci ITS, ACT, CHS-1, TUB2, GAPDH, and ApMat of the *C. gloeosporioides* complex. Values of Bayesian posterior probability (BPP) and support values of Bootstrap (BS) (1000 replicates) are provided at the nodes. Branches that are unsupported with BS or BPP are denoted by –. *Colletotrichum truncatum* CBS 151.35 is treated as an outgroup. The sequences obtained in the current study are indicated in black boldface. The novel species are indicated in blue.

**Figure 3 jof-09-00705-f003:**
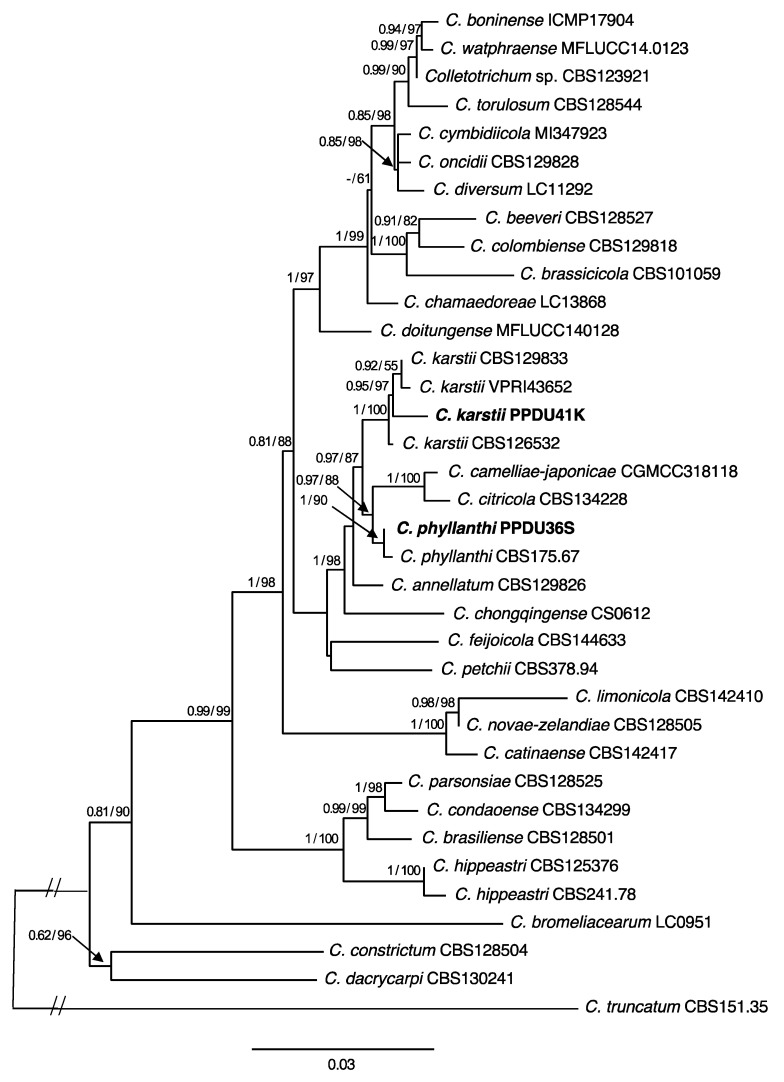
Maximum likelihood tree obtained through heuristic searches of the four loci ITS, ACT, CHS-1, and TUB2 sequences of the *C. boninense* complex. Values of Bayesian posterior probability (BPP) and support values of Bootstrap (BS) (1000 replicates) are provided at the nodes. Branches that are unsupported with BPP or BS are denoted with –. *Colletotrichum truncatum* CBS 151.35 is treated as an outgroup. The sequences obtained in the current study are indicated in black boldface.

**Figure 4 jof-09-00705-f004:**
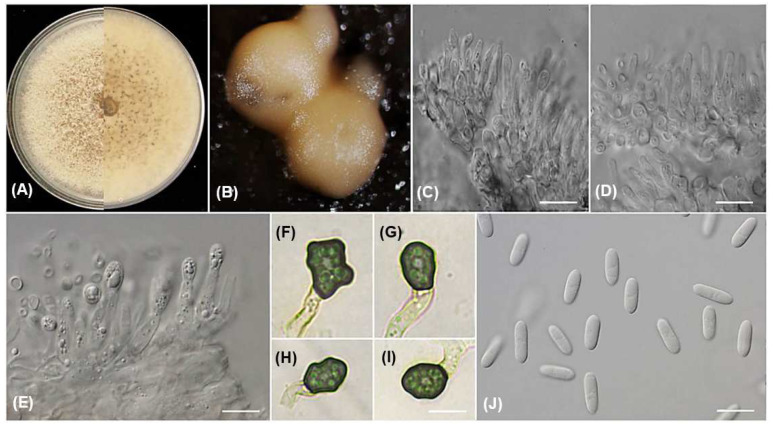
*Colletotrichum saudianum* (from ex-holotype strain PPDU38H). Colony morphology (**A**); pinkish orange masses of conidia releases from acervuli (**B**); hyaline conidiophores (**C**–**E**); appressoria (**F**–**I**); hyaline conidia (**J**). - Scale bars; (**C**–**J**) = 10 µm.

**Figure 5 jof-09-00705-f005:**
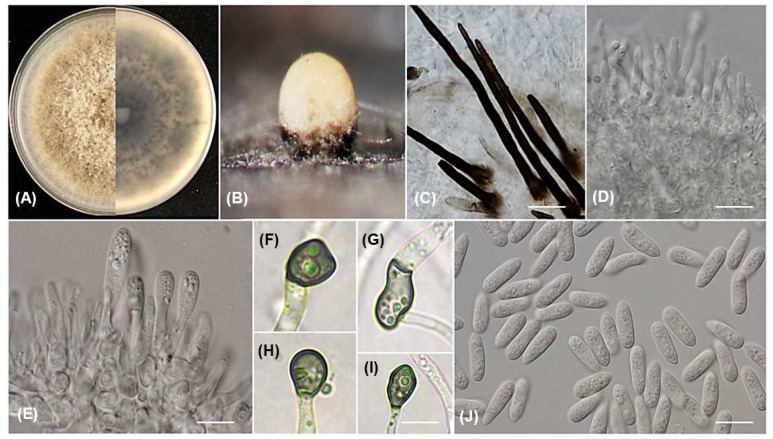
*Colletotrichum coffeae-arabicae* (from ex-holotype strain PPDU26B). Colony morphology (**A**); orange masses of conidia releases from acervuli (**B**); seta (**C**); hyaline conidiophores (**D**,**E**); appressoria (**F**–**I**); hyaline conidia with guttules (**J**). - Scale bars; (**C**–**J**) = 10 µm.

**Figure 6 jof-09-00705-f006:**
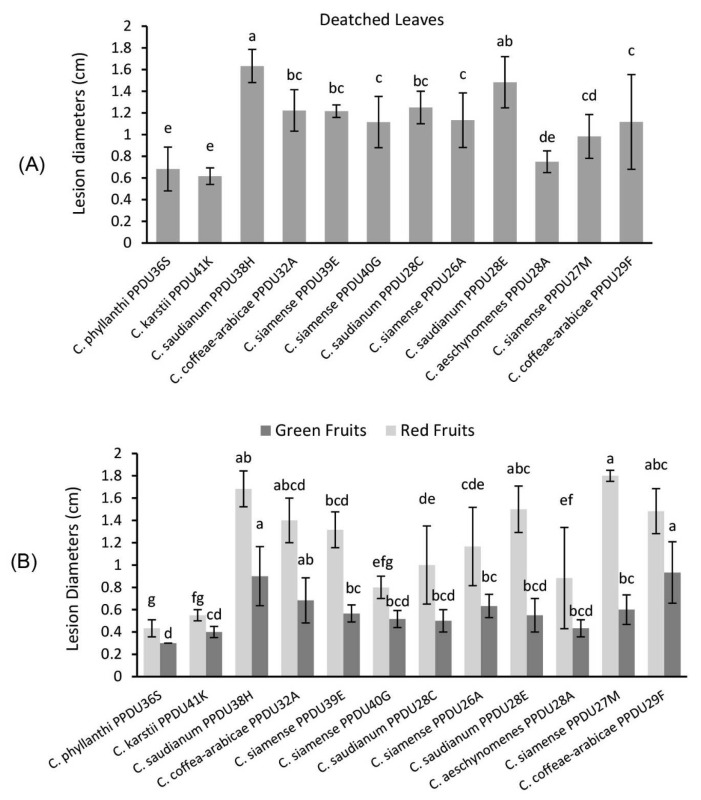
Lesions diameters *(y*-axis) released from 12 *Colletotrichum* isolates (*x*-axis) inoculated on detached coffee leaves (**A**), red and green fruit (**B**) after 10 days of incubation at 25 °C. Each isolate’s values represent the mean of six replicates ± (SD). Means designated with similar letters in these columns did not vary significantly according to the LSD test (*p* < 0.05).

**Figure 7 jof-09-00705-f007:**
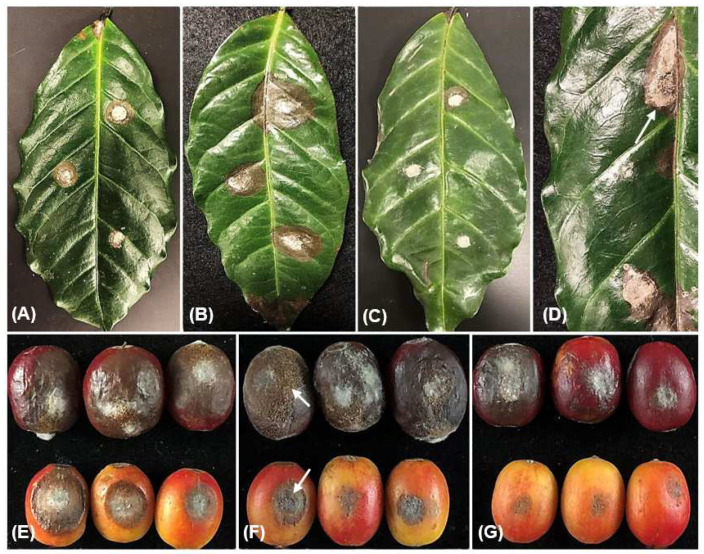
Symptoms reproduced by tested *Colletotrichum* species on detached coffee leaves (**A**–**D**); necrotic lesions developed on red and green fruits after 8 days of incubation at 25 °C (**E**,**F**); small lesions developed by the *C. krastii* PPDU41K showing the weakness of the fungus to reproduce the symptoms observed in the field (**C**,**G**); orange masses of conidia released from semi-immersed acervuli (arrows) observed on the necrotic tissues of leaves and red fruit produced by the virulent isolate of *C. saudianum* PPDU38H (**D**,**F**).

**Table 1 jof-09-00705-t001:** Geographical sites of surveyed coffee plantations in four regions in the southwest of Saudi Arabia.

District	No. of Farms	Longitude (E)	Latitude (N)	Altitude (m)
Jazan	1	43°8′19.9″	17°22′14.3″	785
	2	43°8′20.4″	17°22′22.8″	803
	3	43°8′20.4″	17°22′27.9″	812
	4	43°8′19.9″	17°22′14.3″	1043
	5	43°8′34.9″	17°17′13.5″	861
Asir	6	42°24′39″	18°9′41″	1880
	7	42°22′10″	18°11′43″	1360
	8	42°23′3″	18°11′32″	1500
	9	42°38′3″	18°13′17″	2120
	10	40°18′45″	18°11′21″	1396
	11	42°19′8″	18°12′45″	1510
	12	42°6′4″	18°49′38″	1660
	13	42°5′57″	18°49′32″	1580
	14	42°4′17″	19°9′30″	1320
	15	43°10′47″	17°40′46″	1200
	16	43°10′50″	17°40′50″	1210
Najran	17	44°10′20″	17°29′5″	1290
	18	44°3′36″	17°26′30″	1340
Al Baha	19	41°25′55.1″	19°47′27.5″	1100
	20	41°21′35″	19°45′1.3″	1084
	21	41°22′36.1″	19°43′35.3″	1258
	22	41°21′16.5″	19°45′36″	1204
	23	41°26′25.7″	20°2′9.8″	2187

**Table 2 jof-09-00705-t002:** A list of primers utilized in the current study for PCR amplification and sequencing.

Locus	Product Name	Primer	Sequence (5′–3′)	Reference
ITS	Internal transcribed spacer	ITS-1F	CTT GGT CAT TTA GAG GAA GTA A	[35]
		ITS-4R	TCC TCC GCT TAT TGA TAT GC	
ACT	Actin	ACT-512F	ATG TGC AAG GCC GGT TTC GC	[36]
		ACT-783R	TAC GAG TCC TTC TGG CCC AT	
CHS-1	Chitin synthase	CHS-79F	TGG GGC AAG GAT GCT TGG AAG AAG	[36]
		CHS-345R	TGG AAG AAC CAT CTG TGA GAG TTG	
GAPDH	Glyceraldehyde-3-phosphate dehydrogenase	GDF	GCC GTC AAC GAC CCC TTC ATT GA	[39]
		GDR	GGG TGG AGT CGT ACT TGA GCA TGT	
TUB2	β-Tubulin 2	T1F	AAC ATG CGT GAG ATT GTA AGT	[37]
		Bt2bR	ACC CTC AGT GTA GTG ACC CTT GGC	[38]
ApMat	Mat1–2	AMF1	TCATTCTACGTATGTGCCCG	[29]
		AMR1	CCAGAAATACACCGAACTTGC	

**Table 3 jof-09-00705-t003:** A list of sequences of *C. gloeosporioides* and *C. boninense* species complexes retrieved from the GenBank and the obtained sequences in this study.

Species Identity	Culture No.	Host	Country	GenBank Accession Numbers					
				ITS	ACT	TUB2	CHS-1	GAPDH	ApMat
*C. aenigma*	ICMP 18608 *	*Persea americana*	Israel	JX010244	JX009443	JX010389	JX009774	JX010044	KM360143
*C. aeschynomenes*	ICMP 17673; ATCC 201874 *	*Aeschynomene virginica*	USA	JX010176	JX009483	JX010392	JX009799	JX009930	KM360145
** *C. aeschynomenes* **	**PPDU28A**	** *Coffea arabica* **	**Saudi Arabia**	**OR048775**	**OR050686**	**OR050783**	**OR050738**	**OR050756**	**OR050711**
*C. alatae*	ICMP 17919 *	*Dioscorea alata*	India	JX010190	JX009471	JX010383	JX009837	JX009990	KC888932
*C. alienum*	ICMP 12071 *	*Malus domestica*	New Zealand	JX010251	JX009572	JX010411	JX009882	JX010028	KM360144
*C. analogum*	YMF 1.06943	Unknown	China	OK030860	OK513599	OK513629	OK513559	OK513663	-
*C. annellatum*	CBS 129826 *	*Hevea brasiliensis*	Colombia	JQ005222	JQ005570	JQ005656	JQ005396	-	-
*C. aotearoa*	ICMP 18537 *	*Coprosma* sp.	New Zealand	JX010205	JX009564	JX010420	JX009853	JX010005	KC888930
*C. arecicola*	CGMCC 3.19667	*Areca catechu*	China	MK914635	MK935374	MK935498	MK935541	MK935455	MK935413
*C. artocarpicola*	MFLUCC 18–1167 *	*Artocarpus heterophyllus*	Thailand	MN415991	MN435570	MN435567	MN435569	MN435568	-
*C. asianum*	ICMP 18580; CBS 130418 *	*Coffea arabica*	Thailand	FJ972612	JX009584	JX010406	JX009867	JX010053	FR718814
*C. australianum*	VPRI 43074; UMC001	*Citrus reticulata*	Australia	MG572137	MK473452	MG572148	MW091986	MG572126	MG572170
*C. australianum*	VPRI 43075; UMC002 *	*Citrus sinensis*	Australia	MG572138	MN442109	MG572149	MW091987	MG572127	MG572171
*C. beeveri*	CBS 128527 *	*Brachyglottis repanda*	New Zealand	JQ005171	JQ005519	JQ005605	JQ005345	-	-
*C. boninense*	ICMP 17904; CBS 123755 *	*Crinum asiaticum* var. *sinicum*	Japan	JQ005153	JQ005501	JQ005588	JQ005327	-	-
*C. brasiliense*	CBS 128501 *	*Passiflora edulis*	Brazil	JQ005235	JQ005583	JQ005669	JQ005409	-	-
*C. brassicicola*	CBS 101059	*Brassica oleracea* var. *gemmifera*	New Zealand	JQ005172	JQ005520	JQ005606	JQ005346	-	-
*C. bromeliacearum*	LC0951	Bromeliad	China	MZ595832	MZ664130	MZ673956	MZ799267	-	-
*C. camelliae*	ICMP 10643 *	*Camellia williamsii*	United Kingdom	JX010224	JX009540	JX010436	JX009891	JX009908	KJ954625
*C. camelliae-japonicae*	CGMCC 3.18118 *, LC6416	*Camellia japonica*	China	KX853165	KX893576	KX893580	MZ799271	-	-
*C. cangyuanense*	YMF1.05001	Unknown	China	OK030864	OK513603	OK513633	OK513563	OK513667	
*C. catinaense*	CBS 142417; CPC 27978 *	*Citrus reticulata*	Italy	KY856400	KY855971	KY856482	KY856136	-	-
*C. chamaedoreae*	LC13868, NN052885	*Chamaedorea erumpens*	China	MZ595890	MZ664188	MZ674008	MZ799274	-	-
*C. changpingense*	MFLUCC 15-0022	*Fragaria ananassa*	China	KP683152	KP683093	KP852490	KP852449	KP852469	-
*C. chongqingense*	CS0612	*Camellia sinensis*	China	MG602060	MT976107	MG602044	MT976117	-	-
*C. chrysophilum*	CMM4268 *, CMM 4352	*Musa* sp.	Brazil	KX094252	KX093982	KX094285	KX094083	KX094183	KX094326
*C. cigarro*	ICMP 18534	*Kunzea ericoides*	New Zealand	JX010227	JX009473	JX010427	JX009765	JX009904	HE655657
*C. citricola*	CBS 134228 *	*Citrus unchiu*	China	KC293576	KC293616	KC293656	KY856140	-	-
*C. clidemiae*	ICMP 18658 *	*Clidemia hirta*	USA	JX010265	JX009537	JX010438	JX009877	JX009989	KC888929
*C. cobbittiense*	BRIP 66219	*Cordyline fruticosa*	Australia	MH087016	MH094134	MH094137	MH094135	MH094133	-
** *C. coffeae-arabicae* **	**PPDU26B**	** *Coffea arabica* **	**Saudi Arabia**	**OR048779**	**OR050690**	**OR050787**	**OR050742**	**OR050760**	**OR050715**
** *C. coffeae-arabicae* **	**PPDU27D**	** *Coffea arabica* **	**Saudi Arabia**	**OR048777**	**OR050688**	**OR050785**	**OR050740**	**OR050758**	**OR050713**
** *C. coffeae-arabicae* **	**PPDU29F**	** *Coffea arabica* **	**Saudi Arabia**	**OR048768**	**OR050679**	**OR050776**	**OR050731**	**OR050749**	**OR050704**
** *C. coffeae-arabicae* **	**PPDU32A**	** *Coffea arabica* **	**Saudi Arabia**	**OR048764**	**OR050675**	**OR050772**	**OR050727**	**OR050745**	**OR050700**
*C. colombiense*	CBS 129818 *	unknown	Colombia	JQ005174	JQ005522	JQ005608	JQ005348	-	-
*C. condaoense*	CBS 134299	*Ipomoea pescaprae*	Vietnam	MH229914	-	MH229923	MH229926	-	-
*C. conoides*	CAUG17; MYL24	*Actinidia deliciosa*	China	KY995389	KY995510	KY995473	KY995436	KY995340	MG198007
*C. constrictum*	CBS 128504	*Citrus limon*	New Zealand	JQ005238	JQ005586	JQ005672	JQ005412	-	-
*C. cordylinicola*	MFLUCC 090551; ICMP 18579 *	*Cordyline fruticosa*	Thailand	JX010226	HM470235	JX010440	JX009864	JX009975	JQ899274
*C. cymbidiicola*	IMI 347923 *	*Cymbidium* sp.	Australia	JQ005166	JQ005514	JQ005600	JQ005340	-	-
*C. dacrycarpi*	CBS 130241 *	Unknown	New Zealand	JQ005236	JQ005584	JQ005670	JQ005410	-	-
*C. dimorphum*	YMF1.07309	Unknown	China	OK030867	OK513606	OK513636	OK513566	OK513670	-
*C. diversum*	LC11292, CQ775	*Philodendron selloum*	China	MZ595844	MZ664142	MZ673965	MZ799272	-	-
*C. doitungense*	MFLUCC 14-0128	*Dendrobium* sp.	Thailand	MF448524	MH376385	MH351277	-	-	-
*C. dracaenigenum*	MFLUCC 19-0430	*Dracaena* sp.	Thailand	MN921250	MT313686	-	MT215575	MT215577	-
*C. endophyticum*	CAUG28; YTJB1	*Capsicum* sp.	China	KP145441	KP145329	KP145469	KP145385	KP145413	MH305548
*C. feijoicola*	CBS 144633, CPC 34245	*Acca sellowiana*	Portugal	MK876413	MK876466	MK876507	MK876471	-	-
*C. fructicola*	ICMP 18581; CBS 130416 *	*Coffea arabica*	Thailand	JX010165	FJ907426	JX010405	JX009866	JX010033	JQ807838
*C. fructicola*	VPRI 43079; UMC006	*Citrus reticulata*	Australia	MG572142	MK473454	MG572153	MW091991	MG572131	MG572175
*C. fructivorum*	CBS 133125 *	*Vaccinium macrocarpon*	USA	JX145145	MZ664126	JX145196	MZ799259	MZ664047	JX145300
*C. gloeosporioides*	IMI 356878; ICMP 17821; CBS 112999 *	*Citrus sinensis*	Italy	JX010152	JX009531	JX010445	JX009818	JX010056	JQ807843
*C. gloeosporioides*	VPRI 43076; UMC003	*Citrus sinensis*	Australia	MG572139	MN442110	MG572150	MW091988	MG572128	MG572172
*C. gloeosporioides*	VPRI 10312; A01-10312	*Citrus sinensis*	Australia	MK469996	MK470086	MK470050	MW091972	MK470014	MK470068
*C. gracile*	YMF1.06939	Unknown	China	OK030868	OK513607	OK513637	OK513567	OK513671	-
*C. grevilleae*	CBS 132879 *	*Grevillea* sp.	Italy	KC297078	KC296941	KC297102	KC296987	KC297010	-
*C. grossum*	CGMCC3.17614T; CAUG7; INIFAT 4145	*Capsicum* sp.	China	KP890165	KP890141	KP890171	KP890153	KP890159	MG826119
*C. hebeiense*	MFLUCC13-0726 *	*Vitis vinifera*	China	KF156863	KF377532	KF288975	KF289008	KF377495	KF377562
*C. hederiicola*	MFLU 15-0689	*Hedera helix*	Italy	MN631384	MN635795		MN635794	ON971378	-
*C. helleniense*	CPC 26844; CBS 142418; CBS 142419	*Poncirus trifoliata*	Greece	KY856446	KY856019	KY856528	KY856186	KY856270	MW368907
*C. henanense*	LC3030; CGMCC 3.17354; LF238 *	*Camellia sinensis*	China	KJ955109	KM023257	KJ955257	MZ799256	KJ954810	KJ954524
*C. hippeastri*	CBS 125376 *	*Hippeastrum vittatum*	China	JQ005231	JQ005579	JQ005665	JQ005405	-	-
*C. hippeastri*	CBS 241.78	*Hippeastrum vittatum*	China	JX010293	JX009485	JQ005666	JX009838	-	-
*C. horii*	ICMP 10492 *	*Diospyros kaki*	Japan	GQ329690	JX009438	JX010450	JX009752	GQ329681	JQ807840
*C. hystricis*	CPC 28153; CBS 142411 *	*Citrus hystrix*	Italy	KY856450	KY856023	KY856532	KY856190	KY856274	-
*C. jiangxiense*	LF687 *, CGMCC 3.17361	*Camellia sinensis*	China	KJ955201	KJ954471	KJ955348	MZ799257	KJ954902	KJ954607
*C. kahawae*	IMI 319418; ICMP 17816 *	*Coffea arabica*	Kenya	JX010231	JX009452	JX010444	JX009813	JX010012	JQ894579
*C. karstii*	CBS 126532	*Citrus* sp.	South Africa	JQ005209	JQ005557	JQ005643	JQ005383	-	-
*C. karstii*	CBS 129833	*Musa* sp.	Mexico	JQ005175	JQ005523	JQ005609	JQ005349	-	-
*C. karstii*	VPRI 43652; UMC016	*Citrus sinensis*	Australia	MW081179	MW081187	MW081183	MW081191	-	-
** *C. karstii* **	**PPDU41K**	** *Coffea arabica* **	**Saudi Arabia**	**OR048754**	**OR050665**	**OR050762**	**OR050717**	**-**	**-**
*C. limonicola*	CBS 142410; CPC 31141 *	*Citrus limon*	Malta	KY856472	KY856045	KY856554	KY856213	-	-
*C. makassarense*	CBS 143664, CPC 28612, CPC 28556	*Capsicum annuum*	Indonesia	MH728812	MH781477	MH846560	MH805847	MH728821	MH728831
*C. musae*	ICMP 19119; CBS 116870 *	*Musa* sp.	USA	JX010146	JX009433	HQ596280	JX009896	JX010050	KC888926
*C. nanhuaense*	YMF1.04993	Unknown	China	OK030870	OK513609	OK513639	OK513569	OK513673	-
*C. novae-zelandiae*	CBS 128505 *	*Capsicum annuum*	New Zealand	JQ005228	JQ005576	JQ005662	JQ005402	-	-
*C. noveboracense*	AFKH109	*Malus domestica*	USA	MN646685	MN640565	MN640569		MN640567	MN640564
*C. nullisetosum*	YMF1.06946	Unknown	China	OK030872	OK513611	OK513641	OK513571	OK513675	
*C. nupharicola*	ICMP 18187 *	*Nuphar polysepala*	USA	JX010187	JX009437	JX010398	JX009835	JX009972	JX145319
*C. oblongisporum*	YMF1.06938	Unknown	China	OK030874		OK513643	OK513573	OK513677	-
*C. oncidii*	CBS 129828 *	*Oncidium* sp.	Germany	JQ005169	JQ005517	JQ005603	JQ005343	-	-
*C. pandanicola*	MFLUCC 17-0571	Pandanaceae	Thailand	MG646967	MG646938	MG646926	MG646931	MG646934	-
*C. pandanicola*	SAUCC200204	Unknown	China	MW786641	MW883694	MW888969	MW883685	MW846239	-
*C. pandanicola*	SAUCC201152	Unknown	China	MW786746	MW883702	MW888977	MW883693	MW876478	-
*C. parsonsiae*	CBS 128525 *	*Parsonsia capsularis*	New Zealand	JQ005233	JQ005581	JQ005667	JQ005407	-	-
*C. parvisporum*	YMF1.06942	Unknown	China	OK030876	OK513613	OK513645	OK513575	OK513679	-
*C. perseae*	CBS 141365 *, GA100, GA 170	*Persea americana*	Israel	KX620308	KX620145	KX620341	MZ799260	KX620242	KX620180
*C. petchii*	CBS 378.94 *	*Dracaena marginata*	Italy	JQ005223	JQ005571	JQ005657	JQ005397	-	-
*C. phyllanthi*	CBS 175.67 *	*Phyllanthus acidus*	India	JQ005221	JQ005569	JQ005655	JQ005395	-	-
** *C. phyllanthi* **	**PPDU36S**	** *Coffea arabica* **	**Saudi Arabia**	**OR048762**	**OR050673**	**OR050770**	**OR050725**	**-**	**-**
*C. proteae*	CBS 132882 *	*Protea* sp.	South Africa	KC297079	KC296940	KC297101	KC296986	KC297009	-
*C. pseudotheobromicola*	MFLUCC 18–1602	*Prunus avium*	China	MH817395	MH853681	MH853684	MH853678	MH853675	-
*C. psidii*	ICMP 19120 *	*Psidium* sp.	Italy	JX010219	JX009515	JX010443	JX009901	JX009967	KC888931
*C. queenslandicum*	ICMP 1778 *	*Carica papaya*	Australia	JX010276	JX009447	JX010414	JX009899	JX009934	KC888928
*C. rhexiae*	Coll1026, CBS 133134 *	*Rhexia virginica*	USA	JX145128	MZ664127	JX145179	MZ799258	MZ664046	JX145290
*C. salsolae*	ICMP 19051 *	*Salsola tragus*	Hungary	JX010242	JX009562	JX010403	JX009863	JX009916	KC888925
** *C. saudianum* **	**PPDU28C**	** *Coffea arabica* **	**Saudi Arabia**	**OR048774**	**OR050685**	**OR050782**	**OR050737**	**OR050755**	**OR050710**
** *C. saudianum* **	**PPDU28E**	** *Coffea arabica* **	**Saudi Arabia**	**OR048773**	**OR050684**	**OR050781**	**OR050736**	**OR050754**	**OR050709**
** *C. saudianum* **	**PPDU28J**	** *Coffea arabica* **	**Saudi Arabia**	**OR048772**	**OR050683**	**OR050780**	**OR050735**	**OR050753**	**OR050708**
** *C. saudianum* **	**PPDU28L**	** *Coffea arabica* **	**Saudi Arabia**	**OR048771**	**OR050682**	**OR050779**	**OR050734**	**OR050752**	**OR050707**
** *C. saudianum* **	**PPDU29A**	** *Coffea arabica* **	**Saudi Arabia**	**OR048770**	**OR050681**	**OR050778**	**OR050733**	**OR050751**	**OR050706**
** *C. saudianum* **	**PPDU29B**	** *Coffea arabica* **	**Saudi Arabia**	**OR048769**	**OR050680**	**OR050777**	**OR050732**	**OR050750**	**OR050705**
** *C. saudianum* **	**PPDU31I**	** *Coffea arabica* **	**Saudi Arabia**	**OR048766**	**OR050677**	**OR050774**	**OR050729**	**OR050747**	**OR050702**
** *C. saudianum* **	**PPDU31M**	** *Coffea arabica* **	**Saudi Arabia**	**OR048765**	**OR050676**	**OR050773**	**OR050728**	**OR050746**	**OR050701**
** *C. saudianum* **	**PPDU38B**	** *Coffea arabica* **	**Saudi Arabia**	**OR048761**	**OR050672**	**OR050769**	**OR050724**	**-**	**OR050698**
** *C. saudianum* **	**PPDU38F**	** *Coffea arabica* **	**Saudi Arabia**	**OR048760**	**OR050671**	**OR050768**	**OR050723**	**-**	**OR050697**
** *C. saudianum* **	**PPDU38H***	** *Coffea arabica* **	**Saudi Arabia**	**OR048759**	**OR050670**	**OR050767**	**OR050722**	**-**	**OR050696**
** *C. saudianum* **	**PPDU38I**	** *Coffea arabica* **	**Saudi Arabia**	**OR048758**	**OR050669**	**OR050766**	**OR050721**	**-**	**OR050695**
*C. siamense*	VPRI 43077; UMC004	*Citrus limon*	Australia	MG572140	MK473453	MG572151	MW091989	MG572129	MG572173
*C. siamense*	CPC 30209, UOM 13	*Capsicum annuum*	Indonesia	MH707471	MH781464	MH846547	MH805834	MH707452	MH713897
*C. siamense*	CPC 30210, UOM14	*Capsicum annuum*	Indonesia	MH707472	MH781465	MH846548	MH805835	MH707453	MH713896
*C. siamense*	CPC 30212, UOM16	*Capsicum annuum*	Indonesia	MH707474	MH781467	MH846550	MH805837	MH707455	MH713894
*C. siamense*	CPC 30221, UOM25	*Capsicum annuum*	Thailand	MH707475	MH781468	MH846551	MH805838	MH707456	MH713893
*C. siamense*	CPC 30222, UOM26	*Capsicum annuum*	Thailand	MH707476	MH781469	MH846552	MH805839	MH707457	MH713892
*C. siamense*	CPC 30223, UOM27	*Capsicum annuum*	Indonesia	MH707477	MH781470	MH846553	MH805840	MH707458	MH713891
*C. siamense*	ICMP 18578 CBS 130417 *	*Coffea arabica*	Thailand	JX010171	FJ907423	JX010404	JX009865	JX009924	JQ899289
*C. siamense*	BRIP 54270b; VPRI 43029; A10-43029	*Citrus australasica*	Australia	MK469995	MK470085	MK470049	MW091971	MK470013	MK470067
*C. siamense* syn. *C. endomangiferae*	CMM 3814a	*Mangifera indica*	Brazil	KC702994	KC702922	KM404170	KC598113	KC702955	KJ155453
*C. siamense* syn. *C. hymenocallidis*	CBS 125378, ICMP 18642, LC0043a	*Hymenocallis americana*	China	JX010278	JX009441	JX010410	GQ856730	JX010019	JQ899283
*C. siamense* syn. *C. hymenocallidis*	CBS 112983, CPC 2291	*Protea cynaroides*	Zimbabwe	KC297065	KC296929	KC297100	KC296984	KC297007	KP703761
*C. siamense* syn. *C. hymenocallidis*	CBS 113199. CPC 2290	*Protea cynaroides*	Zimbabwe	KC297066	KC296930	KC297090	KC296985	KC297008	KP703763
*C. siamense* syn. *C. hymenocallidis*	CBS 116868	*Protea cynaroides*	Zimbabwe	KC566815	KC566961	KP703429	KC566382	KC566669	KP703764
*C. siamense* syn. *C. jasmini-sambac*	CBS 130420; ICMP 19118	*Jasminum sambac*	Viet Nam	HM131511	HM131507	JX010415	JX009895	HM131497	JQ807841
** *C. siamense* **	**PPDU26A**	** *Coffea arabica* **	**Saudi Arabia**	**OR048780**	**OR050691**	**OR050788**	**OR050743**	**OR050761**	**OR050716**
** *C. siamense* **	**PPDU27B**	** *Coffea arabica* **	**Saudi Arabia**	**OR048778**	**OR050689**	**OR050786**	**OR050741**	**OR050759**	**OR050714**
** *C. siamense* **	**PPDU27M**	** *Coffea arabica* **	**Saudi Arabia**	**OR048776**	**OR050687**	**OR050784**	**OR050739**	**OR050757**	**OR050712**
** *C. siamense* **	**PPDU29H**	** *Coffea arabica* **	**Saudi Arabia**	**OR048767**	**OR050678**	**OR050775**	**OR050730**	**OR050748**	**OR050703**
** *C. siamense* **	**PPDU32B**	** *Coffea arabica* **	**Saudi Arabia**	**OR048763**	**OR050674**	**OR050771**	**OR050726**	**OR050744**	**OR050699**
** *C. siamense* **	**PPDU39D**	** *Coffea arabica* **	**Saudi Arabia**	**OR048757**	**OR050668**	**OR050765**	**OR050720**	**-**	**OR050694**
** *C. siamense* **	**PPDU39E**	** *Coffea arabica* **	**Saudi Arabia**	**OR048756**	**OR050667**	**OR050764**	**OR050719**	**-**	**OR050693**
** *C. siamense* **	**PPDU40G**	** *Coffea arabica* **	**Saudi Arabia**	**OR048755**	**OR050666**	**OR050763**	**OR050718**	**-**	**OR050692**
*C. subhenanense*	YMF1.06865	Unknown	China	OK030883	OK513618	OK513647	OK513581	OK513684	-
*C. syzygicola*	DNCL021; MFLUCC 10-0624 *, DU-2013c	*Syzygium samarangense*	Thailand	KF242094	KF157801	KF254880	KJ947226	KF242156	KP743473
*C. tainanense*	UOM 1119, Coll 1290	*Capsicum annuum*	Taiwan	MH728805	MH781487	MH846570	MH805857	MH728819	MH728824
*C. tainanense*	CBS 143666, CPC30245,	*Capsicum annuum*	Taiwan	MH728818	MH781475	MH846558	MH805845	MH728823	MH728836
*C. temperatum*	CBS 133122 *	*Vaccinium macrocarpon*	USA	JX145159	MZ664125	JX145211	MZ799254	MZ664045	JX145298
*C. theobromicola*	ICMP 18649; CBS 124945 *	*Theobroma cacao*	Panama	JX010294	JX009444	JX010447	JX009869	JX010006	KC790726
*C. ti*	ICMP 4832 *	*Cordyline* sp.	New Zealand	JX010269	JX009520	JX010442	JX009898	JX009952	KM360146
*C. torulosum*	CBS 128544 *	*Solanum melongena*	New Zealand	JQ005164	JQ005512	JQ005598	JQ005338	-	-
*C. tropicale*	ICMP 18653; CBS 124949 *	*Theobroma cacao*	Panama	JX010264	JX009489	JX010407	JX009870	JX010007	KC790728
*C. truncatum*	CBS 151.35 *	*Phaseolus lunatus*	USA	GU227862	GU227960	GU228156	GU228352	-	-
*C. truncatum*	CBS 151.35 *	*Phaseolus lunatus*	USA	GU227862	GU227960	GU228156	GU228352	-	-
*C. viniferum*	GZAAS 5.08601; GC9	*Vitis vinifera*	China	JN412804	JN412795	JN412813	MW684718	JN412798	MT648530
*C. watphraense*	MFLUCC 14-0123	*Dendrobium* sp.	Thailand	MF448523	MH376384	MH351276	-	-	-
*C. wuxiense*	CGMCC 3.17894 *	*Camellia sinensis*	China	KU251591	KU251672	KU252200	KU251939	KU252045	KU251722
*C. xanthorrhoeae*	BRIP 45094; ICMP 17903; CBS 127831 *	*Xanthorrhoea* sp.	Australia	JX010261	JX009478	JX010448	JX009823	JX009927	KC790689
*C. xishuangbannaense*	MFLUCC 19-0107	*Magnolia liliifera*	China	MW346469	MW652294	-	MW660832	MW537586	-
*C. yuanjiangensis*	YMF1.04996	Unknown	China	OK030885	OK513620	OK513649	OK513583	OK513686	-
*C. yulongense*	CFCC 50818	*Vaccinium dunalianum*	China	MH751507	MH777394	MK108987	MH793605	MK108986	-
*Colletotrichum* sp.	CBS 123921, MAFF 238642	*Dendrobium kingianum*	Japan	JQ005163	JQ005511	JQ005597	JQ005337	-	-

* Represent ex-type isolates. The isolates obtained in this study are boldfaced.

## Data Availability

All the data related to this study are mentioned in the manuscript and Appendix A.

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
