# Peer review of "Multi-Locus Phylogenetic Analysis Revealed the Association of Six Colletotrichum Species with Anthracnose Disease of Coffee (Coffea arabica L.) in Saudi Arabia"

_jof, 2023, doi:10.3390/jof9070705_

Round 1
Reviewer 1 Report
The manuscript entitled "Multi-locus Phylogenetic Analysis Revealed the Association of 2 Six Colletotrichum Species with Anthracnose Disease of Coffee 3 (Coffea arabica L.) in Saudi Arabia" has been reviewed. The manuscript was well prepared. However, there are some advices as following:
1. Lines 18-20, 122-124, the sentences should be rewritten.
2. Line 189, How to do the surface sterilization for the fruit? How many times repeated the pathogenicity test?
3. The gene abbreviations (GAPDH or Gapdh?) should be uniformed in the manuscript. All should be carefully checked.
4. In the Figures, (A), (B),… there is special signal after each of them. Can you delete it?
5. The information (leave, fruit,…; isolation month; location) of the 27 isolates should be given in the manuscript.
6. There are some revisions marked in the attached file.

There are some revisions marked in the attached file.
Author Response
Dear Professor,
We greatly appreciate all the critiques and comments from you. Those comments are extremely helpful for us to improving our paper, and they provide valuable guidance for our future study. According to these comments, we have carefully improved our manuscript.
We have made improvments in throughout the whole manuscript as recommended, and these revisions made to the manuscript were marked up using the “Track Changes” function to be easily viewed by the editors and reviewers.
Yours sincerely,
Associ. Prof. Dr. Ahmed M. Ismail
Reviewer 2 Report
Arabica coffee crops are threatened worldwide by the fungal pathogen Colletotrichum. The authors identified the strains of Colletotrichum that cause anthracnose disease in coffee in Saudi Arabia. The strains were identified using molecular method based on sequence data base of fungi. Authors sequenced PCR products obtained using ITS region, chitin synthase I, GAPDH, actin, β-tubulin and mating type genes. The 27 isolates were classified into two Colletotrichum species complexes: C. gloeosporioides and C. boninense. Results of the molecular analysis were then used in the phylogenetic trees construction and detailed classification of the isolates.
The Authors identified 6 species of Colletotrichum, four of which, C. aeschynomenes, C. siamense, C. karstii and C. phyllanthi, had previously been identified, and two were new: Colletotrichum saudianum and Colletotrichum coffeae-arabicae. The last two strains belong to the C. gloeosporioides complex.
The Authors characterized the two new strains of Colletotrichum and these characteristics were sent to the MycoBank.
Are there any differences between these two new strains? It's hard to pick out these differences from the text.
The pathogenicity test revealed differences in the virulence of the isolates. It seems that the new strains, especially C. saudianum PPDU38H was the most virulent and caused the biggest sessions on leaves and on the green and red fruits. The next isolate of C. saudianum PPDU28C caused much smaller lesions on the green and red fruits. The Authors did not discussed these differences. What are the reasons for such large differences in the virulence of the two isolates of C. saudianum.? It is not known whether the isolates were collected from the same places from which part of the coffee plant they came. Was C. saudianum found in the same places as, for example, C. karstii and C. phyllanthi? Were the most aggressive isolates found together with benign isolates?
What is the different between green and red fruits? Why are red fruits more intensively attacked by the pathogens? Should be explained in the results
In the Discussion The Authors discuss the suitability of particular sequences for strain identification. They also summarize the identification results.
The most represented isolates were identified as C. saudianum (12 isolates from 27). The Authors pointed at the fact that more than one species can colonize the same host. C. karstii and C. phyllanthi were classified as of little importance in coffee infections. These species were previously found on coffee in Vietnam and Mexico. The Authors confirmed that all isolates were pathogenic although in various degree.
Figure 6 shows the severity of the infection syndromes, but an additional important parameter was mentioned in the discussion that C. saudianum and C. siamense developed lesions 3 days earlier than C. karstii and C. phyllanthi. This is an important result because it shows the variability in the virulence of the strains. Authors should mention this in the results
In this study, the Authors identified two new strains of Colletotrichum. They noted the importance of identifying the pathogen for the safety of the coffee industry. However, they did not propose how this knowledge could help fight the disease.
Line 432 …., five were already known, …… should be four
Table 3 should be moved to the supplementary data.
Author Response

(The authors gave the same response as above.)

Round 2
Reviewer 2 Report
Most of my suggestions have been accepted. The removal of Table 3 from the main text to the supplementary data did not make much difference, although this is a very large Table and the data is not as relevant to the manuscript. I leave it to the authors' decision.